DATA RELEASE

# Annotation of chitin biosynthesis genes in *Diaphorina citri*, the Asian citrus psyllid

Sherry Miller[1,2], Teresa D. Shippy[1], Blessy Tamayo[3], Prashant S. Hosmani[4], Mirella Flores-Gonzalez[4], Lukas A. Mueller[4], Wayne B. Hunter[5], Susan J. Brown[1], Tom D'Elia[3] and Surya Saha[4,6,*]

1 Division of Biology, Kansas State University, Manhattan, KS 66506, USA
2 Allen County Community College, Burlingame, KS 66413, USA
3 Indian River State College, Fort Pierce, FL 34981, USA
4 Boyce Thompson Institute, Ithaca, NY 14853, USA
5 USDA-ARS, U.S. Horticultural Research Laboratory, Fort Pierce, FL 34945, USA
6 Animal and Comparative Biomedical Sciences, University of Arizona, Tucson, AZ 85721, USA

## ABSTRACT

The polysaccharide chitin is critical for the formation of many insect structures, including the exoskeleton, and is required for normal development. Here we report the annotation of three genes from the chitin synthesis pathway in the Asian citrus psyllid, *Diaphorina citri* (Hemiptera: Liviidae), the vector of Huanglongbing (citrus greening disease). Most insects have two chitin synthase (CHS) genes but, like other hemipterans, *D. citri* has only one. In contrast, *D. citri* is unusual among insects in having two UDP-N-acetylglucosamine pyrophosphorylase (UAP) genes. One of the *D. citri* UAP genes is broadly expressed, while the other is expressed predominantly in males. Our work helps pave the way for potential utilization of these genes as pest control targets to reduce the spread of Huanglongbing.

**Subjects** Genetics and Genomics, Animal Genetics, Bioinformatics

**Submitted:** 19 December 2020

* Corresponding author. E-mail: suryasaha@cornell.edu

Preprint submitted at https://doi.org/10.1101/2020.09.22.309211

## DATA DESCRIPTION

### Introduction

Chitin is a polysaccharide that is essential for insect development. It is crucial in the development of the insect cuticle and exoskeleton, the peritrophic membrane of the midgut of some insects, and other structures such as the trachea, wing hinges and eggshell [1]. Because chitin is essential for insect development but is not found in mammals, the enzymes involved in its synthesis are considered attractive targets for pest control. The biosynthetic pathway for chitin begins with the hexosamine pathway, in which simple sugars, such as glucose, trehalose and glycogen, are converted into UDP-N-acetylglucosamine (UDP-GlcNAc). The final step in the hexosamine pathway is catalyzed by the enzyme UDP-N-acetylglucosamine pyrophosphorylase (UAP) [1]. UDP-GlcNAc is then converted to chitin by enzymes known as chitin synthases (CHS) [1].

### Context

Here we report the annotation of the CHS and UAP genes in genome version 3 (v3) of the Asian citrus psyllid, *Diaphorina citri* (Hemiptera: Liviidae; NCBI:txid121845), the vector for

⊕ Annotating genes in *Diaphorina citri* genome version 3  ▾

Teresa D Shippy[1], S Miller[2], C Massimino[3], C Vosburg [Indian River State College[4], PS Hosmani[5], M Flores-Gonzalez[5], LA Mueller[5], WB Hunter[6], JB Benoit[7], SJ Brown[1], T D'elia[3], Surya Saha[5]

[1]Kansas State University; [2]Kansas State University, Allen County Community College; [3]Indian River State College; [4]Indian River State College, The Pennsylvania State University; [5]Boyce Thompson Institute; [6]USDA-ARS U.S. Horticultural Research Laboratory; [7]University of Cincinnati

Dec 17, 2020

⊟ Run

☆ Bookmark

⚑ Copy / Fork

2 | Works for me     dx.doi.org/10.17504/protocols.io.bniimcce

D. citri annotation

👤 Teresa Shippy
Kansas State University

**Figure 1.** Annotation protocol for psyllid genome curation [16]. https://www.protocols.io/widgets/doi?uri=dx.doi.org/10.17504/protocols.io.bniimcce

**Table 1.** Chitin synthase and UAP ortholog number in select insects.

| | *Drosophila melanogaster* | *Anopheles gambiae* | *Aedes aegypti* | *Tribolium castaneum* | *Apis mellifera* | *Nasonia vitripennis* | *Acyrthosiphon pisum* | *Bemisia tabaci* | *Diaphorina citri* |
|---|---|---|---|---|---|---|---|---|---|
| CHS1/A | 1 | 1 | 1 | 1 | 1 | 1 | 1 | 1 | 1 |
| CHS2/B | 1 | 1 | 1 | 1 | 1 | 1 | 0 | 0 | 0 |
| UAP | 1 | 1 | 1 | 2 | 1 | 1 | 1 | 1 | 2 |

Gene counts are taken from published reports [3, 5, 6] or determined from genome data [7–14]. *D. citri* numbers are based on annotation of genome v3.

the bacterium that causes Huanglongbing (citrus greening disease). The *D. citri* v3 genome is a chromosome-level assembly with a 40.5-megabase pair (Mb) scaffold N50 value, and 88.3% complete Benchmarking Universal Single-Copy Orthologs (BUSCO) [2]. However, due to heterogeneity of the sequenced psyllids, the genome has numerous false duplications of varying sizes, ranging from multiple adjacent genes to partial exons. As with all genomes, computationally annotated models provide a starting point, but often require manual correction.

We identified and manually annotated one CHS gene and two UAP genes in the *D. citri* genome v3. Although most insects have two CHS genes [3, 4] (Table 1), the presence of a single CHS gene is consistent with reports from other hemipteran genomes [5]. In contrast, *D. citri* seems to be unusual in that it has two UAP genes. Available RNA-seq data indicate that one of the *D. citri* UAP genes is broadly expressed, while the other is expressed predominantly in males. Our manual annotation of these chitin biosynthesis genes provides more accurate information for the design of future experiments involving these genes.

## METHODS

*D. citri* genes in genome v3 [2] were identified by BLAST (NCBI BLAST, RRID:SCR_004870) analysis of *D. citri* sequences with insect CHS and UAP orthologs. Reciprocal BLAST of the National Center for Biotechnology Information (NCBI) non-redundant protein database [15] was used to confirm orthology. Manual annotation of genes was performed in Apollo (Apollo, RRID:SCR_001936; v2.1.0) using RNA-seq reads, Iso-seq transcripts and *de novo*-assembled transcripts as evidence. A more detailed description of the annotation workflow is available via protocols.io (Figure 1) [16].



**Table 2.** Orthologs used in phylogenetic analysis.

| Species | Accession | Name in NCBI | Name in Tree |
|---|---|---|---|
| *Tribolium castaneum* | NP_001034491.1 | chitin synthase 1 | Tc CHS1 |
| *Anopheles gambiae* | XP_321336.5 | AGAP001748-PA | Ag CHS1 |
| *Apis mellifera* | XP_016770736.1 | PREDICTED: uncharacterized protein LOC412215 isoform X1 | Am LOC412215 |
| *Nasonia vitripennis* | XP_008215129.1 | PREDICTED: uncharacterized protein LOC100118280 isoform X1 | Nv LOC100118280 |
| *Acyrthosiphon pisum* | XP_003247517.1 | PREDICTED: uncharacterized protein LOC100162079 | Ap LOC100162079 |
| *Bemisia tabaci* | XP_018916997.1 | PREDICTED: uncharacterized protein LOC109044007 isoform X1 | Bt LOC109044007 |
| *Drosophila melanogaster* | NP_524233.1 | krotzkopf verkehrt, isoform A | Dm krotzkopf verkehrt |
| *Manduca sexta* | AAL38051.2 | chitin synthase | Ms CHS1 |
| *Spodoptera exigua* | AAZ03545.1 | chitin synthase A | Se CHSA |
| *Tribolium castaneum* | NP_001034492.1 | chitin synthase 2 | Tc CHS2 |
| *Manduca sexta* | AAX20091.1 | chitin synthase 2 | Ms CHS2 |
| *Spodoptera exigua* | ABI96087.1 | chitin synthase B | Se CHSB |
| *Drosophila melanogaster* | NP_524209.3 | chitin synthase 2 | Dm CHS2 |
| *Anopheles gambiae* | XP_321951.2 | AGAP001205-PA | Ag CHS2 |
| *Apis mellifera* | XP_016767448.1 | chitin synthase chs-2 | Am CHS-2 |
| *Nasonia vitripennis* | XP_008215122.2 | chitin synthase chs-2 | Nv CHS-2 |
| *Drosophila melanogaster* | NP_001285673.1 | mummy, isoform D | Dm Mummy |
| *Anopheles gambiae* | XP_317600.4 | AGAP007889-PA | Ag UAP |
| *Aedes aegypti* | EAT47260.1 | AAEL001627-PA | Aa UAP |
| *Bombyx mori* | NP_001296486.1 | UDP-N-acetylhexosamine pyrophosphorylase-like protein 1 | Bm UAP |
| *Tribolium castaneum* | NP_001164533.1 | UDP-N-acetylglucosamine pyrophosphorylase 1 | Tc UAP1 |
| *Tribolium castaneum* | NP_001164534.1 | UDP-N-acetylglucosamine pyrophosphorylase 2 | Tc UAP2 |
| *Apis mellifera* | XP_624349.1 | UDP-N-acetylhexosamine pyrophosphorylase | Am UAP |
| *Nasonia vitripennis* | XP_001602623.1 | UDP-N-acetylhexosamine pyrophosphorylase | Nv UAP |
| *Acyrthosiphon pisum* | XP_001944680.1 | UDP-N-acetylhexosamine pyrophosphorylase | Ap UAP |
| *Bemisia tabaci* | XP_018902053.1 | PREDICTED: UDP-N-acetylhexosamine pyrophosphorylase | Bt UAP |
| *Locusta migratoria* | AGN56418.1 | UDP N-acetylglucosamine pyrophosphorylases 1 | Lm UAP1 |
| *Locusta migratoria* | AGN56419.1 | UDP N-acetylglucosamine pyrophosphorylases 2 | Lm UAP2 |
| *Leptinotarsa decemlineata* | XP_023024177.1 | UDP-N-acetylhexosamine pyrophosphorylase-like | Ld UAP1 |
| *Leptinotarsa decemlineata* | XP_023022882.1 | UDP-N-acetylhexosamine pyrophosphorylase-like protein 1 | Ld UAP2 |

Species, NCBI Accession numbers, full names and abbreviated names used in phylogenetic trees are listed for all orthologs included in phylogenetic analyses (Figures 2, 3).

Multiple alignments of the predicted *D. citri* proteins and their insect homologs were performed using MUSCLE (RRID:SCR_011812) [17] or CLUSTALW (RRID:SCR_002909) [18] within MEGAX (MEGA software, RRID:SCR_000667), as specified in each figure legend. Phylogenetic trees were constructed using full-length protein sequences in MEGAX. Orthologs used in tree construction are listed in Table 2. Gene expression levels (Table 3) were obtained from the Citrus Greening Expression Network [19] and visualized using Excel (Microsoft Excel, RRID:SCR_016137) and the pheatmap package (pheatmap, RRID:SCR_016418) in R (R Project for Statistical Computing, RRID:SCR_001905) [20, 21].

## DATA VALIDATION AND QUALITY CONTROL
### Chitin synthases
Chitin synthases are the only enzymes in the chitin biosynthetic pathway that act specifically in the synthesis of chitin. This makes them an attractive, insect-specific target for RNA interference (RNAi)-based insecticides. The two *CHS* genes found in most



**Table 3.** TPM expression values.

| Gene/Transcript name | CHS-RA | CHS-RB | UAP1 | UAP2 |
|---|---|---|---|---|
| Gene ID | Dcitr04g09970.1.1 | Dcitr04g09970.1.2 | Dcitr08g04630.1.1 | Dcitr05g05060.1.1 |
| Egg *Citrus macrophylla C*Las− Whole body | 29.67 | 5.79 | 76.03 | 0.28 |
| Nymph *Citrus medica C*Las+ Low infection Whole body | 28.07 | 50.83 | 53.3 | 3.04 |
| Nymph *Citrus sinensis C*Las+ High infection Whole body | 18.9 | 57.96 | 48.58 | 2.89 |
| Nymph *C. sinensis C*Las− Whole body | 10.8 | 57.65 | 43.84 | 2.25 |
| Nymph *C. macrophylla C*Las− Whole body | 51.71 | 20.61 | 22.3 | 2.3 |
| Nymph *Citrus* spp. *C*Las− Whole body | 21.04 | 0 | 24.12 | 0.17 |
| Nymph *Citrus* spp. *C*Las+ Whole body | 16.14 | 0 | 112.11 | 3.96 |
| Adult *C. medica C*Las− Gut | 0.21 | 0 | 16.28 | 1.41 |
| Adult *C. medica C*Las+ Gut | 0.04 | 0.01 | 15.36 | 0.53 |
| Adult *C. medica C*Las+ High infection Whole body | 8.52 | 2 | 18.82 | 24.16 |
| Adult *C. medica C*Las+ Low infection Whole body | 6.67 | 7.11 | 22.09 | 26.83 |
| Adult *C. medica C*Las− Whole body | 14.39 | 22.71 | 25.51 | 17.25 |
| Adult *C. macrophylla C*Las− Whole body | 0.51 | 0 | 26.1 | 48.95 |
| Adult *Citrus* spp. *C*Las− Whole body | 0.19 | 0 | 12.56 | 40.68 |
| Adult *Citrus* spp. *C*Las+ Whole body | 0.41 | 0 | 29.15 | 18.13 |
| Adult *Citrus* spp. *C*Las− midgut | 0.15 | 0 | 28.82 | 1.12 |
| Adult *Citrus* spp. *C*Las+ midgut | 0.69 | 0 | 20.8 | 5.57 |
| Adult *Citrus reticulata C*Las− Female abdomen | 0.44 | 0 | 72.64 | 0.5 |
| Adult *C. reticulata C*Las− Female antennae | 0.65 | 0.09 | 70.19 | 1.59 |
| Adult *C. reticulata C*Las− Female head | 0.73 | 0 | 73.58 | 0.09 |
| Adult *C. reticulata C*Las− Female leg | 0.41 | 0 | 109.73 | 0 |
| Adult *C. reticulata C*Las− Female terminal abdomen | 1.01 | 0 | 149.58 | 1.03 |
| Adult *C. reticulata C*Las− Female thorax | 0.49 | 0 | 40.29 | 0.28 |
| Adult *C. reticulata C*Las− Male abdomen | 0.35 | 0 | 50.21 | 34.24 |
| Adult *C. reticulata C*Las− Male antennae | 1.17 | 0.13 | 56.8 | 10.87 |
| Adult *C. reticulata C*Las− Male head | 0.77 | 0 | 59.63 | 0.29 |
| Adult *C. reticulata C*Las− Male leg | 0.12 | 0 | 55.29 | 12.29 |
| Adult *C. reticulata C*Las− Male terminal abdomen | 0.96 | 0 | 92.77 | 19.86 |
| Adult *C. reticulata C*Las− Male thorax | 0.25 | 0 | 31.74 | 2.03 |
| Adult *C. reticulata C*Las− Female antennae [22] | 1.41 | 0.44 | 27.94 | 0.03 |
| Adult *C. reticulata C*Las− Female terminal abdomen [22] | 0.32 | 0 | 44.29 | 0.99 |
| Adult *C. reticulata C*Las− Male antennae [22] | 3.68 | 0.44 | 27.89 | 5.05 |
| Adult *C. reticulata C*Las− Male terminal abdomen [22] | 0.59 | 0 | 38.01 | 39.26 |

CHS-RA: Chitin synthase-RA; CHS-RB: Chitin synthase-RB; *C*Las: *Candidatus* Liberibacter asiaticus; UAP1: UDP-N-acetylglucosamine pyrophosphorylase 1; UAP2: UDP-N-acetylglucosamine pyrophosphorylase 2. TPM values for annotated chitin biosynthesis genes from available RNA-seq experiments. All data is publicly available and was obtained from the Citrus Greening Expression Network (CGEN) [19]. For each sample, information on developmental stage, food source, *C*Las infection status and tissue are provided in the first column.

holometabolous insects have distinct functions. *CHS1*, also referred to as *CHSA*, produces the chitin essential for proper cuticle development [4, 23, 24]. *CHS2*, also referred to as *CHSB*, is not required for cuticle development, but is instead essential for proper development of the gut peritrophic membrane [4, 23, 24]. RNAi knockdown of either *CHS* gene is lethal in holometabolous insects [25–28].

Previous searches of the *Acyrthosiphon pisum*, *Nilaparvata lugens* and *Rhodnius prolixus* genomes identified *CHS1* but not *CHS2*, suggesting that *CHS2* has probably been lost in the hemipteran lineage [5]. Loss of the chitin synthase gene required for peritrophic membrane development is not particularly surprising, since hemipterans do not have peritrophic membranes [5, 29]. Lu *et al.* [30] identified a *D. citri CHS* gene that clustered with other hemipteran *CHS* genes and was expressed at high levels in most adult body tissues, but at low levels in midgut, as would be expected for a *CHS1* gene. Two groups have shown that

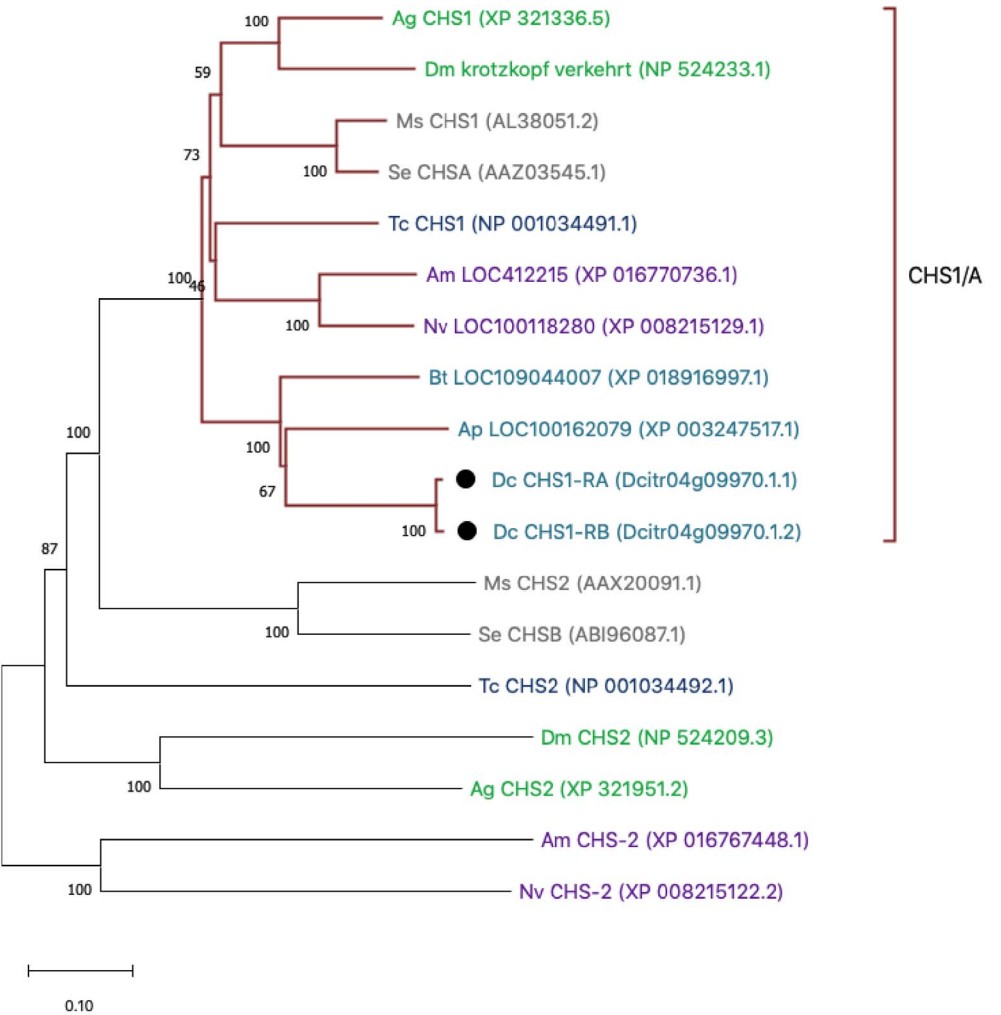

**Figure 2.** Phylogenetic analysis of insect CHS proteins. Species represented are *Drosophila melanogaster* (Dm), *Anopheles gambiae* (Ag), *Tribolium castnaeum* (Tc), *Manduca sexta* (Ms), *Spodoptera exigua* (Se), *Apis mellifera* (Am), *Nasonia vitripennis* (Nv), *Acyrthosiphon pisum* (Ap), *Bemisia tabaci* (Bt) and *Diaphorina citri* (Dc). MUSCLE (RRID:SCR_011812) [31] software was used to perform multiple sequence alignments of full-length protein sequences and the tree was constructed with MEGA X (RRID:SCR_000667) [32] software using the neighbor-joining method with 100 bootstrap replications. The maroon clade shows monophyletic clustering of CHS1/A genes. With the exception of *D. citri* (denoted by black circles), only one isoform per species is depicted. Taxon name color represents insect order: Diptera (green), Coleoptera (navy), Hymenoptera (purple), Lepidoptera (gray), and Hemiptera (teal).

RNAi knockdown of *CHS* in *D. citri* causes increased lethality [30, 33], supporting the idea that this gene is a good target for pest control.

Our searches of the *D. citri* v3 genome revealed the previously described *CHS* gene, but no additional chitin synthase orthologs (Table 1). Transcriptomic evidence supports the existence of two *CHS* isoforms (Table 4) that differ only in the use of one alternative exon and produce proteins with slightly different C-termini. Similar isoforms of *CHS1/A* have been described in other insects [3, 34, 35]. Both isoforms of *D. citri* CHS cluster in a monophyletic clade with CHS1 proteins from other insects (Figure 2), so we have named this gene *CHS1*. We retrieved expression data for both isoforms of CHS1 from the Citrus

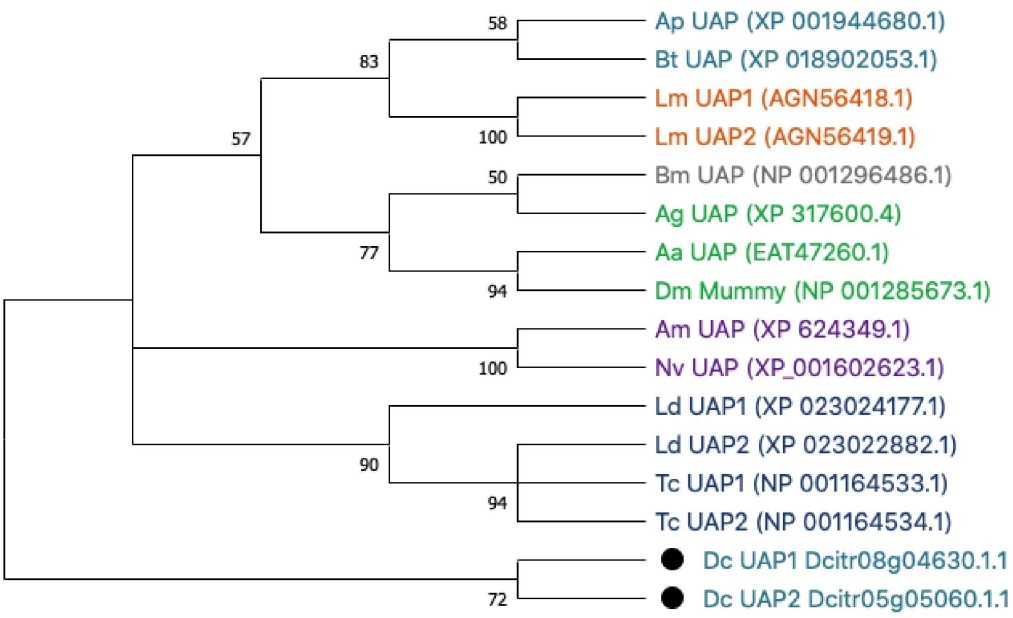

**Figure 3.** Phylogenetic analysis of representative insect UAP orthologs. Species shown are *Drosophila melanogaster* (Dm), *Anopheles gambiae* (Ag), *Aedes aegypti* (Aa), *Bombyx mori* (Bm), *Tribolium castaneum* (Tc), *Leptinotarsa decemlineata* (Ld), *Apis mellifera* (Am), *Nasonia vitripennis* (Nv), *Locusta migratoria* (Lm), *Acyrthosiphon pisum* (Ap), *Bemisia tabaci* (Bt) and *Diaphorina citri* (Dc and black circles). ClustalW software was used to perform the multiple sequence alignment of full-length protein sequences and a bootstrap consensus tree was constructed with MEGA X software using the neighbor-joining method with 100 bootstrap replications. Colors denote insect orders: Hemiptera (teal), Orthoptera (orange), Lepidoptera (gray), Diptera (green), Hymenoptera (purple) and Coleoptera (navy).

**Table 4.** Annotated *D. citri* orthologs of chitin biosynthesis genes.

| Gene/Isoform | OGSv3 ID | Gene model | Evidence supporting annotation | | | |
|---|---|---|---|---|---|---|
| | | Complete | MCOT | Iso-seq | RNA-seq | Ortholog |
| CHS1 | Dcitr04g09970.1.1 | X | MCOT15276.0.CT | X | X | X |
| | Dcitr04g09970.1.2 | | MCOT13830.0.CO | | | |
| UAP1 | Dcitr08g04630.1.1 | X | | X | | X |
| UAP2 | Dcitr05g05060.1.1 | X | | X | X | X |

MCOT: MAKER (**MAKER, RRID:**SCR_005309), Cufflinks (**Cufflinks, RRID:**SCR_014597), Oases (**Oases, RRID:**SCR_011896), Trinity (**Trinity, RRID:**SCR_013048) pipeline. Each manually annotated gene has been assigned an OGSv3 gene identifier and denoted as a partial or complete model based on available evidence. Evidence types used for manual annotation are shown for each gene. A description of the various evidence sources and their strengths and weaknesses is included in our online protocol [16].

Greening Expression Network (CGEN), which contains RNA-seq data sets for various life stages and tissues [19]. Data from whole body samples indicate that CHS1 is expressed at all life stages, but is most highly expressed in juvenile stages (Figure 4).

Our manual annotation of *CHS1* corrects several errors that were present in the previous computationally predicted annotation for *D. citri CHS* (XP_017303059). Changes to the model include the addition of formerly missing sequence and the removal of artifactually duplicated regions. Domain analysis with TMHMM Server (TMHMM Server, RRID:SCR_014935, v2.0) indicates that the corrected CHS1-RA and CHS1-RB proteins have 15 transmembrane helices, as is typical for insect CHS proteins, rather than the 14 that were reported for the earlier version of the protein [30].

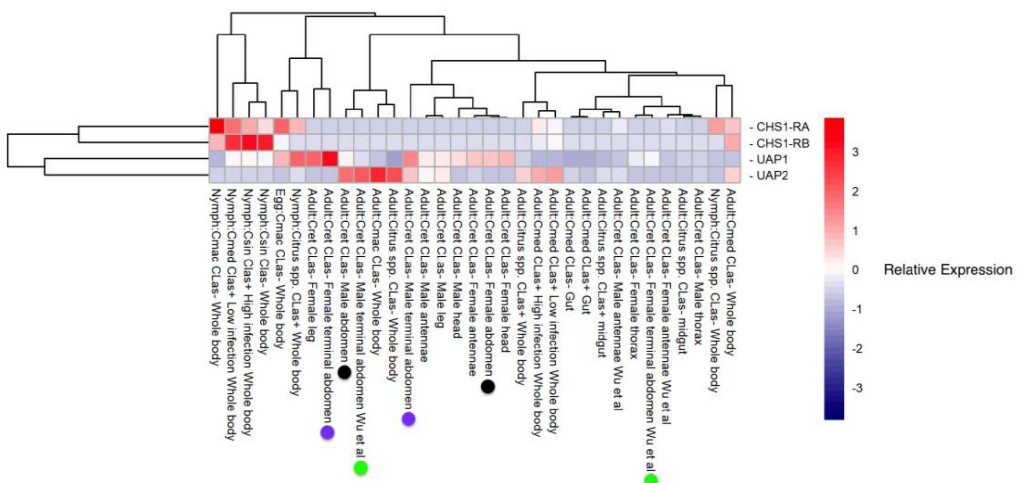

**Figure 4.** Heatmap representation of chitin biosynthesis gene expression levels in various RNA-seq datasets. Expression levels were obtained as transcripts per million (TPM) from the Citrus Greening Expression Network [19] and the heatmap was scaled by row. For ease of comparison, colored circles denote pairs of male and female abdominal tissue samples from the same experiments.

## UDP-N-acetylglucosamine pyrophosphorylase (UAP)

In addition to its role in chitin synthesis, UAP is involved in the modification of other carbohydrates, sphingolipids and proteins. In *Drosophila*, mutants of *UAP* (also called *mummy*, *cabrio* and *cystic*) have defects in tracheal development, dorsal closure, eye development and nervous system function [36–38]. Some of these developmental defects are caused by disruption of the chitin synthesis pathway, while others appear to be caused by effects on other glycoproteins. For example, defects in embryonic dorsal closure have been linked to a role for UAP in regulation of Decapentaplegic signaling [6].

Most insects appear to have a single *UAP* gene (Table 1) [39]. However, a few insects, including *T. castaneum, Locusta migratoria* and *Leptinotarsa decemlineata* have two UAP genes [39–41]. Comparison of the *T. castaneum* and *L. migratoria* gene pairs indicates that they arose through separate, relatively recent lineage-specific gene duplications [40]. RNAi experiments in *T. castaneum* showed that UAP1 is involved in the biosynthesis of chitin both in the cuticle and the peritrophic membrane, while UAP2 is important for the modification of other macromolecules [39]. In *L. migratoria, LmUAP1* knockdown caused lethality and defects consistent with disruption of chitin biosynthesis, while *LmUAP2* knockdown did not increase lethality and produced no visible effects [40].

In the *D. citri* v3 genome, we identified two *UAP* genes located on different chromosome-length scaffolds. The proteins encoded by these apparent paralogs share 50% identity, distributed throughout the length of the proteins (Figure 5), which is similar to the level of identity shared with UAP orthologs from closely related insect species. Amino acid residues known to be important for substrate binding in the human UAP ortholog and conserved in the *T. castaneum* UAP proteins [39] are also well conserved in the *D. citri* UAP proteins (Figure 5). Phylogenetic analysis (Figure 3) suggests that the two genes represent a lineage-specific duplication. Surprisingly, the *D. citri* UAP proteins do not cluster with the other hemipteran UAP proteins; instead, they appear as an outgroup to all the other insect UAP proteins. This suggests that the *D. citri UAP* genes are diverging rather rapidly. We have

```
Dc_UAP1      MVDINEGKDLLTSCNQQ-------------HLLKYWDEINDEEKSILLNE---IKQLNIP
Dc_UAP2      -MDSTKSQDIFTSENSEIDPNKGRTLQKSLELEPNENVSKDKELNILLKDDVSLMNMDIF
              :*  .:.:*::** *.:                  *     :  :*:*  .***::    :  :::*

Dc_UAP1      EACEYFKSANTYSRQAEAQCDSSMKPVPSELYGSAQDTSSDILKSYREIGLQEISEGHVG
Dc_UAP2      KARKYYEEACAVSNAPERFSEVQVFP-PNCLSG-VNTVDASTLGKYRELGLKLISRGDVA
              :*  :*::.*  :  *. .*   .:  .:  * *.  * *  .:  ..:.  *  .***:**:  ** *  *.

Dc_UAP1      VILLAGGQGTRIGVPYPKGMYKVGLPSDKSLFQIQAERIMKLESLAFEQTGKKSIITWFI
Dc_UAP2      VIVLAGGQGTRLGADYPKGMYNIGLPSGKSLFQIQAEKIDKLIEIAKEKFG-SGCLPWFI
              **:*********:*.  ******::****.*********.*  **  .:* *:  *  ..  :.***

Dc_UAP1      MTSESTMEPTKNFFEENKYFGLQKDNVIFFEQGVLPCFTFDGKIIMDSKFKIAKAPDGNG
Dc_UAP2      MTSELTDRPTREYFERNGYFGLDPAHVIFFKQRSMPCFSLSGEILLETRDRVARSPDGHG
              **** *   **.:;** * ****:  :****:*   :***::..*:*:::::.  .:*.:***:*

Dc_UAP1      GIYIALKKKGILSEMEKRGIEYVHVYSVDNILVKVADPVFMGFCVKSQSDCGVKVVEKKL
Dc_UAP2      GLYHALGATGILDTMHTRGIKHIHVYCVDNILVKVGDPTFLGYCVEQGAHCGVKVVEKIT
              *:* **   .***. *  .***:::***.********.**.*:*:**:.  :  ********

Dc_UAP1      PNEGLGVVCVVDGQYKVVEYSEISSKTA-ELRDADGKLTFRAGNICNHFFSTAFLGQIAN
Dc_UAP2      PGESLGVLCNVDGKHKIVEYSELGNCSVFETQDQTGRLKFNLGSICNHYFSLECLQRMVK
              *.*.***:* ***::*:*****:.. :.  *  .*   *.*.*.  *.*****:**   *  .:.:

Dc_UAP1      EHESKLKLHIAKKKIPYIDSKGLKVKPEQPNGIKIEKFIFDVFEFCKNLVVWEVAREHDF
Dc_UAP2      E-DAALKFHMARKKIPCLDEQGISQRPNKPNGIKLEKFLFDAFPLCENLVAWEVTRD-EF
              *  ::  **:*:*.****  :*.:*:.  .*::*****:***:**.*  :*:***.***:*:  :*

Dc_UAP1      SALKNSNAEK-TENPTTCCLALYDLHKSYIEAAGGTVKPDAVGNVVCEISPSVSYDGEGL
Dc_UAP2      SPLKNSPLDSASDNPVTCCQAVHALHARWIETAGGVVVADETGNTVCEIAPRVSYEGEGL
              *.**** ::. ::**.*** *:: **  :**:***.* .* .**.****:* ***:****

Dc_UAP1      KPIVNGNTFESPILLK
Dc_UAP2      EERVKGKVLQTPLLLE
              :   *:*:.:::*:**:
```

**Figure 5.** Alignment of *D. citri* UAP1 and UAP2. Alignment was performed using MUSCLE (MUSCLE, RRID:SCR_011812) [17]. Individual amino acid alignments are denoted as identical (*), highly similar (:) or similar (.). Residues important for substrate binding by human UAP1 and conserved in *T. castaneum* are shaded according to their level of conservation. Identical residues are shaded blue and non-identical (but similar) residues are shaded red. The green shaded residue denotes the position of an alanine important for substrate binding in human UAP1 that is a cysteine in *T. castaneum* and other insects.

named the *D. citri* genes *UAP1* and *UAP2*, but no implication is intended of direct orthology with duplicated UAP genes in other insects.

We compared available expression data from the two *D. citri UAP* genes using CGEN [19]. *D. citri UAP1* is expressed in all tissues and stages examined, although expression levels vary (Figure 4). A few samples (e.g. female terminal abdomen and female leg) show high expression of *UAP1*, but these are single replicate samples that would need further verification. In the case of female terminal abdomen, single replicate data from a separate experiment shows only a moderate level of expression. Interestingly, *D. citri UAP2* appears to show a sexually dimorphic expression pattern. It is expressed at a low-to-moderate level in most male tissues, with highest expression in abdominal samples, but shows little or no expression in the same tissues from females (Figures 4, 6). While these observations are intriguing, the technical difficulty of creating RNA-seq libraries from miniscule amounts of dissected tissue, while maintaining the integrity of the RNA, in addition to the lack of statistical power provided by single replicate samples, mean that the expression data

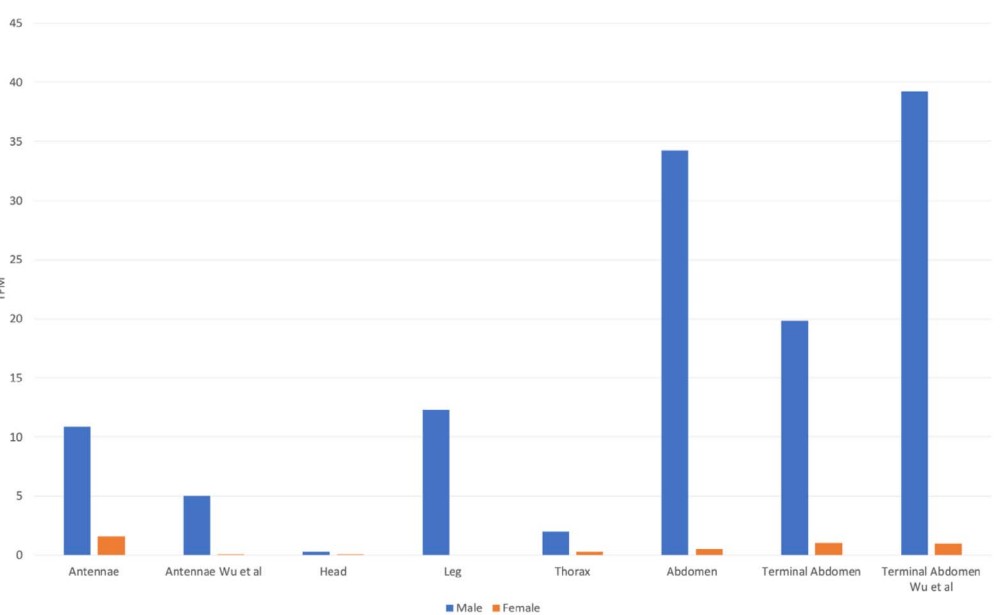

**Figure 6.** Expression levels of *UAP2* in male and female tissues. Expression levels were obtained from the Citrus Greening Expression Network [19]. Tissue types are shown on the *X* axis and expression levels (TPM) on the *Y*-axis. Blue bars denote expression levels in males and orange bars denote expression levels in females (all single replicate data). RNA-seq data from tissues labeled Wu *et al.* were sequenced in [22]. Data for the remaining tissues are from NCBI BioProject PRJNA448935.

currently available should be interpreted with caution. Experimental analysis is outside the scope of this data release, but additional studies of *UAP1* and *UAP2* expression and function in individual males and females will be necessary to verify these results.

## RE-USE POTENTIAL

There is considerable interest in use of the genes described here as targets for pest control. At least two groups have already begun functional studies of the CHS gene in *D. citri*. Our improved annotations will allow more detailed experiments to be performed in the future. For example, isoform-specific RNAi experiments on the *CHS* gene could be designed to determine the function of each transcript variant. The revised gene models will be incorporated into a new official gene set, which will be available for BLAST analysis and expression profiling on the Citrus Greening website [42] and the CGEN [19].

## DATA AVAILABILITY

The *Diaphorina citri* genome assembly, official gene sets, and transcriptome data are accessible via the Citrus Greening website [42]. All accessions for genes used for phylogenetic analysis are provided within this report, and all other data are available in the *GigaScience* GigaDB repository [43].

## EDITOR'S NOTE

This article is one of a series of Data Releases crediting the outputs of a student-focused and community-driven manual annotation project curating gene models and if required, correcting assembly anomalies, for the *Diaphorina citri* genome project [2].

## DECLARATIONS
## LIST OF ABBREVIATIONS

CGEN: Citrus Greening Expression Network; CHS: chitin synthase; *C*Las: *Candidatus Liberibacter asiaticus*; NCBI: National Center for Biotechnology Information; OGS: Official Gene Set; RNAi: RNA interference; TPM: transcripts per million; UAP: UDP-N-acetylglucosamine pyrophosphorylase; UDP-GlcNAc: UDP-N-acetylglucosamine.

## ETHICAL APPROVAL

Not applicable.

## CONSENT FOR PUBLICATION

Not applicable.

## COMPETING INTERESTS

The authors declare that they have no competing interests.

## FUNDING

This work was supported by USDA-NIFA grant 2015-70016-23028, HSI 1300394, 2020-70029-33199 and an Institutional Development Award (IDeA) from the National Institute of General Medical Sciences of the National Institutes of Health under grant number P20GM103418.

## AUTHORS' CONTRIBUTIONS

WBH, SJB, TD and LAM conceptualized the study; TD, SS, TDS and SJB supervised the study; SJB, TD, SS, and LAM contributed to project administration; SM, TDS, and BT conducted investigation; PH, MF-G, and SS contributed to software development; SS, TDS, PH, and MF-G developed methodology; SJB, TD, WBH, and LAM acquired funding; SM and TDS prepared and wrote the original draft; SS, WBH and SJB reviewed and edited the draft.

## ACKNOWLEDGEMENTS

We thank Dr. Josh Benoit for assistance with data visualization.

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
